# From gym to joy: The serial mediation of motor competence and health literacy in Chinese university students' exercise-life satisfaction pathway

Shao-Shuai Ma[1], Zhe Zhu[2], Dongsheng Cai[3], Chen-Xi Li[3], Ya-Xing Li[1], Bo Li[3], Sai Zhu[4], Jiaxian Geng[5]*

1 Physical Education Institute, Shangqiu University, Shangqiu, China, 2 Graduate School of Sports Science, Dongshin University, Naju-si, Korea, 3 Institute of Sports Science, Nantong University, Nantong, China, 4 Nantong Health College of Jiangsu, Nantong, China, 5 Physical Education Institute, Huzhou University, Huzhou, China

* zcx66886688@163.com

## Abstract

### Purpose

To investigate the impact of exercise adherence on life satisfaction among Chinese university students and to explore the mediating roles of motor-skill competence and health literacy. The ultimate goals are to provide a comprehensive understanding of the mechanisms underlying the relationship between physical activity and well-being, and to inform the development of targeted interventions that promote holistic student development.

### Method

A total of 15,031 valid responses were extracted from a national university-student survey database. All variables were assessed with standardized questionnaires. Data were analyzed with SPSS 27.0. Promoting the physical and mental well-being and holistic development of students has become a policy priority that commands national attention, public concern, and sustained governmental focus, and the PROCESS macro v4.0.

### Result

Exercise adherence, life satisfaction, motor-skill competence, and health literacy were positively intercorrelated. Exercise adherence significantly and positively predicted university students' life satisfaction. The indirect effect along Path 1—exercise adherence→motor-skill competence→life satisfaction—was 0.003, accounting for 1.1% of the total effect. The indirect effect along Path 2—exercise adherence→health literacy→life satisfaction—was 0.067, accounting for 26.25% of

**Data availability statement:** Data cannot be shared publicly, because data from this study may contain sensitive patient information. Data are anonymized, however, due to relatively few severe cases, patients could be identified (The Ethics Committee at Nantong University). Therefore, data from this study will be made available for researchers who meet criteria for access to confidential data. Requests may be sent to: ethics@ntu.edu.cn.

**Funding:** The author(s) received no specific funding for this work.

**Competing interests:** The authors have declared that no competing interests exist.

the total effect. The indirect effect along Path 3—exercise adherence→motor-skill competence→health literacy→life satisfaction—was 0.001, accounting for 0.28% of the total effect.

## Conclusion

The study findings reveal that exercise adherence has a positive direct effect on university students' life satisfaction and also exerts an indirect effect through the sequential mediation of motor-skill competence and health literacy. These results provide empirical evidence of the complex interplay between physical activity, skill development, and mental health.

---

## 1. Introduction

Promoting the physical and mental well-being and holistic development of students has become a policy priority that commands national attention, public concern, and sustained governmental focus [1–3]. The eighth National Survey on Students' Physical Fitness and Health, released in 2021, reported a gradual increase in the proportion of students achieving "excellent" or "good" physical-fitness standards; this improvement was most pronounced among junior-high-school students following the reform of the physical-examination admission policy. University students, however, lack an equivalent mandatory physical-fitness assessment comparable to the senior-high-school entrance examination, and both their physical-fitness levels and overall physical capacity have declined [4]. This downward trend in university students' physical health not only signifies a regression in objective physiological indicators but also indirectly erodes their life satisfaction [5,6]. On the path toward the ambitious Healthy China 2030 goals, university students' life satisfaction is pivotal not only for individual growth and happiness but also for social stability and long-term national development [7–10]. Comparisons between the 2022 and 2024 China College Student Satisfaction Surveys indicate that the proportion of students who reported being "relatively satisfied" with campus life rose from 78.6% to 83.4%, suggesting an overall upward trajectory. Yet, most respondents focused on canteen food quality, dormitory hygiene, and administrative services, paying limited attention to their own physical and psychological health. Consequently, while their evaluations of university life improved, their bodies increasingly "expressed dissatisfaction" [11–13]. Against this backdrop, a deeper investigation into the determinants of university students' life satisfaction is urgently warranted.

Life satisfaction, the core cognitive component of subjective well-being, captures individuals' global appraisal of life quality and plays a pivotal role in emotional regulation and psychological adaptation [14–16]. Within this framework, the persistent nature of exercise behavior—commonly termed exercise adherence—has emerged as a key lens through which to examine the psychological benefits of physical activity among university students [17,18]. Exercise adherence is defined as the behavioral process by which an individual engages in physical exercise continuously and

regularly over an extended period, with emphasis on tangible behavioral outcomes [19]. Empirical evidence consistently demonstrates a robust association between physical exercise and life satisfaction, a linkage that appears to be under-pinned by the formation of stable behavioral habits and the cumulative accrual of psychological resources [20–22]. A lon-gitudinal study by Wang revealed that students who maintained exercise participation for more than six months reported significantly greater gains in life satisfaction than their short-term counterparts [23]. Complementing this finding, Li demon-strated that when exercise frequency was sustained at four or more sessions per week. Each session exceeded 45 min, and individuals exhibited pronounced improvements in goal attainment and self-efficacy—psychological attributes that directly enhance life satisfaction [24]. Drawing on the self-determination theory advanced by Edward L. Deci and Richard M. Ryan [25], exercise adherence is posited to foster a positive psychological cycle by satisfying three basic psychological needs: autonomy, through the sense of control derived from regular exercise; competence, via progressive improvements in motor-skill mastery; and relatedness, cultivated within exercise-oriented social networks. The sustained accumulation of such psychological capital equips individuals with greater resilience against external stressors, thereby deepening their positive evaluation of life. Accordingly, we advance Hypothesis 1: Exercise adherence will significantly predict university students' life satisfaction.

Motor-skill competence denotes the integrated capacity for transferable movement patterns that individuals acquire through systematic practice [26]. Gentile's motor-learning theory posits that such mastery requires repeated practice, cumulative experience, and continuous feedback [27]; accordingly, it depends not only on the accrual of repetitions but also on the learner's cognitive reconstruction and adaptive adjustment of movement patterns [28]. Sustained exercise adherence supplies the indispensable practice volume for skill refinement, whereas improved proficiency enhances psychological adaptation via two synergistic pathways—increased self-efficacy and heightened achievement motivation—thereby exerting a positive effect on life satisfaction [6,29,30].

Viewed through the lens of Locke and Latham's goal-setting theory, the progressive mastery of motor skills can be construed as a series of "stepwise goal attainments." Each incremental success satisfies the need for competence posited by self-determination theory, fostering positive self-appraisals of ability that generalize to evaluations in other life domains [31–33]. Moreover, advanced motor-skill competence endows individuals with a sense of bodily control and performance confidence that effectively buffers the psychological impact of stressful events, initiating the chained reaction "exercise adherence → kill improvement → psychological-capital accumulation → optimized life satisfaction" [34]. On this basis, we propose Hypothesis 2: Motor-skill competence mediates the relationship between exercise adherence and life satisfaction.

Health literacy is defined as the capacity to access, understand, and apply health-related information to make appro-priate health decisions [35]. Sustained exercise adherence not only enhances physical fitness but also deepens indi-viduals' understanding of exercise-science principles, nutritional management, and mind–body interactions through repeated health-behavior practice, thereby systematically elevating health-literacy levels [36,37]. In a national sample, Yu observed that university students scored highest on health-knowledge tests yet reported the lowest levels of actual health behavior [38]; consequently, poor dietary choices, physical inactivity, and irregular lifestyles frequently place their health status "on red alert." Li's further analyses revealed that improvements in physical fitness foster scientifically informed exercise awareness and encourage students to priorities their health in daily life; in turn, this deliberate engagement in evidence-based physical activity enhances physical capacity, facilitates academic and occupational functioning, and ulti-mately exerts a positive influence on life satisfaction [39]. Accordingly, we propose Hypothesis 3: Health literacy mediates the association between exercise adherence and life satisfaction.

Motor-skill competence and health literacy are significantly and positively correlated. Sun reported that only 3.86% of university students were classified as having high physical-activity levels, yet 45.65% of these highly active students demonstrated adequate health literacy [40]. Building on this finding, we hypothesize that motor-skill competence may exert an indirect effect on life satisfaction through health literacy, and that exercise adherence may further influence life satisfaction via the serial pathway of motor-skill competence followed by health literacy. We therefore propose Hypothesis

4: Motor-skill competence and health literacy serially mediate the relationship between exercise adherence and university students' life satisfaction.

In summary, to elucidate the underlying mechanisms linking university students' exercise adherence to their life satisfaction, we constructed a theoretical model that specifies the pathways of influence; this model is depicted in **Fig 1**.

## 2. Methods

### 2.1. Participants

The study's sample encompassed multiple provinces and universities, providing broad representation. However, the uneven distribution of participants across gender and grade levels may affect the generalizability of the findings. Specifically, the sample included a higher proportion of female students and students from higher grade levels. This uneven distribution may introduce sampling bias, potentially affecting the representativeness of the results. To address this, we conducted stratified analysis by gender and grade level to control for these differences. We acknowledge this limitation and discuss its potential impact on the study's generalizability in the Discussion section.

We use a large-scale cross-sectional survey study, using questionnaire star software, and the scales used are mature scales that can be applied to college students. **The recruitment period for the study spanned from 08/10/2024 to 09/11/2024, with the subsequent questionnaire survey being conducted from 11/11/2024 to 24/11/2024.** The study protocol for this study received approval from the ethics committee at Nantong University and was documented under approval number 2022(70). Prior to commencing the formal investigations and testing, the researchers obtained informed consent from all the participants involved in the study. Before survey administration, all participants provided written informed consent through digital signature on the online questionnaire platform. The consent form explicitly outlined: (a) research purpose, (b) voluntary participation nature, (c) confidentiality protocols, and (d) data usage terms in compliance with the Declaration of Helsinki. Participants were full-time students enrolled in regular higher-education institutions located in mainland China; the institutional roster followed the Ministry of Education's "National List of Regular Higher-Education Institutions" (current to 20 June 2024). Sampling criteria were:

(a) campuses situated in the provinces of Beijing, Chongqing, Anhui, Gansu, Guizhou, Henan, Heilongjiang, Hubei, Hunan, Jilin, Jiangxi, Inner Mongolia, and Shaanxi;

(b) age 18–22 years;

(c) complete data on the four core variables (exercise adherence, life satisfaction, motor-skill competence, and health literacy).

**Fig 1. Mediational model of variables.**

After listwise deletion, 15,031 valid cases remained. The sample spans multiple provinces and institutional types, ensuring broad representativeness; details are reported in Table 1.

## 2.2. Instruments

### (1)   Exercise Adherence Scale

Exercise adherence was assessed with Gu's 2016 Physical Exercise Adherence Scale [41]. The scale comprises three dimensions—behavioral engagement, effort expenditure, and emotional experience—and contains seven items rated on a five-point Likert scale (1 = strongly disagree, 5 = strongly agree). Higher summed scores indicate greater adherence. In the present sample, the scale demonstrated excellent internal consistency (Cronbach's α = 0.947) [42].

### (2) Life Satisfaction Scale

Life satisfaction was measured with the 13-item Satisfaction with Life Scale, revised by Ed Diener et al. [43]. Each item is rated on a 7-point Likert scale (1 = strongly disagree, 7 = strongly agree). Higher summed scores indicate greater life satisfaction. Following established norms for Chinese university students, scores of 31–35 denote very high satisfaction, 26–30 denote high satisfaction, 21–25 denote moderate satisfaction, 20 denote neutral, 15–20 denote moderate dissatisfaction, 10–14 denote dissatisfaction, and 5–9 denote very high dissatisfaction [44]. In the present sample, the scale's Cronbach's α was 0.78 [45].

### (3) Motor-Skill Competence Scale

Motor-skill competence was assessed based on the "Youth Motor-Skill Proficiency Standards and Testing Methods" published by Chen et al. [46]. Participants were surveyed regarding their proficiency in specific sports skills using the following question [47]: "How many sports skills are you proficient in (e.g., basketball, football, volleyball, martial arts, taekwondo, table tennis, badminton, swimming, gymnastics, dance, ice skating, ice hockey, skipping rope, roller skating, etc.)?" Response options were 0, 1, or 2 or more sports. However, this single-item measure may not fully capture the complexity of motor-skill competence, which involves multiple dimensions such as strength, coordination, and accuracy. Additionally, the reliability and validity of this measure were not established in this study, which may limit the accuracy and meaningfulness of the findings related to motor-skill competence.

### (4)   Health Literacy Scale

Health literacy was measured with the Simplified Health Literacy Scale, revised by Sun et al. in 2024 [48]. The scale comprises three dimensions: health care, disease prevention, and health promotion. It contains nine items rated on a 4-point Likert scale (1 = tough, 2 = difficult, 3 = easy, 4 = very easy). Higher scores indicate better health literacy. The Health Literacy

**Table 1.  Sample distribution table.**

| Variables | | *n* | % |
|---|---|---|---|
| Gender | | | |
| | Male | 6353 | 42.3 |
| | Female | 8678 | 57.7 |
| Grade Level | | | |
| | Freshman | 9386 | 62.4 |
| | Sophomore | 5138 | 34.2 |
| | Junior | 445 | 3.0 |
| | Senior | 62 | 0.4 |

(HL) index is calculated as follows: HL index = (mean score – 1) × (50/3). In the present study, the scale demonstrated excellent internal consistency (Cronbach's α = 0.913) [48]. The scale's construct validity was confirmed through exploratory and confirmatory factor analyses in a Chinese population, with items loading significantly on their respective dimensions [49]. Additionally, the scale showed good convergent validity, with significant correlations with established health behavior measures. These findings, supported by previous research, confirm that the Simplified Health Literacy Scale is a valid and reliable tool for assessing health literacy in China [49].

### 2.4. Statistical methods

Common method bias was first assessed using the Harman single-factor test. The results showed that there were eight factors with eigenvalues greater than 1, and the variance explained by the first factor was 29.107%, which is below the commonly used threshold of 40%. Thus, no significant common method bias was detected in this study. Next, descriptive analysis was conducted to calculate the sample size (n), percentage, chi-square value, Cramer's V coefficient, and differences by gender and grade for motor-skill competence and life satisfaction. Pearson's correlation coefficients (Pearson's r) were then used to examine the associations between variables for correlation analysis. Subsequently, multiple linear regression analysis was performed with exercise adherence as the independent variable, life satisfaction as the dependent variable, and health literacy and motor-skill competence as control variables, producing standardized regression coefficients (β), the coefficient of determination ($R^2$), and significance levels (P values). Finally, a chain mediation model was constructed using the PROCESS macro (Model 6) in SPSS 27.0 to test the serial mediation effect of health literacy and motor-skill competence. The Bootstrap method was employed with 5,000 resamples to calculate the 95% confidence intervals (95% CI) for the total effect, direct effect, and indirect effects. Path significance was determined by whether the 95% CI included zero. A P value of less than 0.01 was considered statistically significant.

## 3. Result

### 3.1. Descriptive results

Before testing the mechanism by which exercise adherence influences life satisfaction, we first described the basic distributions of all variables. Cross-tabulations were generated for the two ordinal variables—motor-skill competence and life satisfaction—reporting sample sizes, percentages, chi-square values, and Cramer's V. Exercise adherence and health literacy, treated as continuous variables, were summarized descriptively; higher scores indicate better status. Table 2 shows the distributions by gender and grade. Overall, 55.4% of students reported being "satisfied" or "very satisfied" with life; this proportion was higher among women (35.4%) than men (32.0%), and the gender difference was significant ($\chi^2 = 92.143$, P < 0.001). Regarding motor-skill competence, 89.7% of students had mastered at least one sport; men (92.7%) outperformed women (87.5%), $\chi^2 = 286.644$, P < 0.001. Across grades, first-year students had the highest proportion with no mastered skill (11.3%), whereas fourth-year students had the highest proportion with two or more skills (67.7%); the grade-level difference was also significant ($\chi^2 = 44.868$, P < 0.001).

### 3.2. Correlation analysis

Correlation analysis (Table 3) showed that all variables were positively and significantly related. Exercise adherence correlated strongly with life satisfaction (r = 0.394, P < 0.01), motor-skill competence (r = 0.259, P < 0.01), and health literacy (r = 0.464, P < 0.01). Life satisfaction had a weak positive association with motor-skill competence (r = 0.120, P < 0.01) and a strong positive association with health literacy (r = 0.367, P < 0.01). Motor-skill competence and health literacy were weakly positively correlated (r = 0.137, P < 0.01).

**Table 2. Summary of descriptive analysis results.**

| Variables | Evaluation Level | Gender | | | | Grade | | | | | | | |
|---|---|---|---|---|---|---|---|---|---|---|---|---|---|
| | | Male(n = 6353) | | Female(n = 8678) | | Freshman | | Sophomore | | Junior | | Senior | |
| | | n | % | n | % | n | % | n | % | n | % | n | % |
| Life Satisfaction | | | | | | | | | | | | | |
| | Very Dissatisfied | 107 | 1.7% | 76 | 0.9% | 101 | 1.1% | 75 | 1.5% | 7 | 1.6% | 0 | 0% |
| | Dissatisfied | 350 | 5.5% | 426 | 4.9% | 501 | 5.3% | 252 | 4.9% | 19 | 4.3% | 4 | 6.5% |
| | Basically Dissatisfied | 1004 | 15.8% | 1533 | 17.7% | 1680 | 17.9% | 801 | 15.6% | 56 | 12.6% | 0 | 0% |
| | Neutral | 977 | 15.4% | 1142 | 13.2% | 1242 | 13.2% | 818 | 15.9% | 53 | 11.9% | 6 | 9.7% |
| | Basically Satisfied | 2033 | 32.0% | 3076 | 35.4% | 3235 | 34.5% | 1703 | 33.1% | 147 | 33.0% | 24 | 38.7% |
| | Satisfied | 1363 | 21.5% | 1937 | 22.3% | 2022 | 21.5% | 1122 | 21.8% | 130 | 29.2% | 26 | 41.9% |
| | Very Satisfied | 519 | 8.2% | 488 | 5.6% | 605 | 6.4% | 367 | 7.1% | 33 | 7.4% | 2 | 3.2% |
| | $\chi^2$ | 92.143 | | | | 81.790 | | | | | | | |
| | P | <.001 | | | | <.001 | | | | | | | |
| | Cramer's V | 0.078 | | | | 0.043 | | | | | | | |
| Motor-skill Competence | | | | | | | | | | | | | |
| | 0item | 466 | 7.3% | 1082 | 12.5% | 1064 | 11.3% | 442 | 8.6% | 36 | 8.1% | 6 | 9.7% |
| | 1item | 1811 | 28.5% | 3193 | 36.8% | 3183 | 33.9% | 1671 | 32.5% | 136 | 30.6% | 14 | 22.6% |
| | ≥2items | 4076 | 64.2% | 4403 | 50.7% | 5139 | 54.8% | 3025 | 58.9% | 273 | 61.3% | 42 | 67.7% |
| | $\chi^2$ | 286.644 | | | | 44.868 | | | | | | | |
| | P | <.001 | | | | <.001 | | | | | | | |
| | Cramer's V | 0.138 | | | | 0.039 | | | | | | | |

**Table 3. Correlation Analysis.**

| | Variables | | Exercise Adherence | Life Satisfaction | Motor-Skill Competence | Health Literacy |
|---|---|---|---|---|---|---|
| Pearson | Exercise Adherence | r | 1 | | | |
| | Life Satisfaction | r | 0.394** | 1 | | |
| | Motor-Skill Competence | r | 0.259** | 0.120** | 1 | |
| | Health Literacy | r | 0.464** | 0.367** | 0.137** | 1 |

Note: ** indicates P<0.01

### 3.3. Mediation analysis

Using hierarchical regression, we examined total, direct, and indirect effects while controlling for gender, grade, and ethnicity. **Table 4** shows that, after these controls, exercise adherence positively predicted life satisfaction (β = 0.253, SE = 0.005, t = 53.695, P < 0.001), confirming the main effect. It also positively predicted motor-skill competence (β = 0.018, SE = 0.001, t = 30.2947, P < 0.001) and health literacy (β = 0.212, SE = 0.004, t = 60.849, P < 0.001). Motor-skill competence positively predicted life satisfaction (β = 0.154, SE = 0.064, t = 2.421, P < 0.05), and health literacy strongly predicted life satisfaction (β = 0.313, SE = 0.011, t = 28.285, P < 0.001). Thus, exercise adherence significantly predicts life satisfaction, supporting H1.

Using bias-corrected percentile bootstrapping and controlling for gender and grade, we examined the serial mediation of motor-skill competence and health literacy in the relationship between exercise adherence and life satisfaction. Results appear in **Table 5**. The total effect of exercise adherence on life satisfaction was 0.253, 95% CI [0.244, 0.263]; the total direct effect was 0.183, 95% CI [0.173, 0.194], accounting for 72.40% of the total; and the total indirect effect was 0.070, 95% CI [0.063, 0.077]. Decomposed effects showed that the pathway via health literacy alone was 0.067, via motor-skill

**Table 4. Hierarchical regression results for the chain mediation effects.**

| Regression equations | | Overall fit indices | | | Regression coefficient significance | | |
|---|---|---|---|---|---|---|---|
| Outcome variable | Predictor variable | R | R² | F | β | SE | t |
| Motor-Skill Competence | | 0.283 | 0.08 | 326.536 | | | |
| | Exercise Adherence | | | | 0.018 | 0.001 | 30.295*** |
| | Gender | | | | −0.121 | 0.011 | −11.073*** |
| | Grade | | | | 0.073 | 0.009 | 7.920*** |
| | Ethnicity | | | | 0.116 | 0.020 | 5.768*** |
| Health Literacy | | 0.464 | 0.216 | 826.202 | | | |
| | Exercise Adherence | | | | 0.212 | 0.004 | 60.849*** |
| | Motor-Skill Competence | | | | 0.120 | 0.047 | 2.575* |
| | Gender | | | | 0.160 | 0.063 | 2.550* |
| | Grade | | | | 0.032 | 0.053 | 0.610 |
| | Ethnicity | | | | −0.061 | 0.115 | −0.528 |
| Life Satisfaction | | 0.453 | 0.205 | 645.376 | | | |
| | Exercise Adherence | | | | 0.183 | 0.005 | 34.703*** |
| | Motor-Skill Competence | | | | 0.154 | 0.064 | 2.421* |
| | Health Literacy | | | | 0.313 | 0.011 | 28.285*** |
| | Gender | | | | 0.781 | 0.085 | 9.159*** |
| | Grade | | | | 0.421 | 0.072 | 5.84*** |
| | Ethnicity | | | | −0.003 | 0.156 | −0.016 |
| Life Satisfaction | | 0.403 | 0.162 | 726.737 | | | |
| | Exercise Adherence | | | | 0.253 | 0.005 | 53.695*** |
| | Gender | | | | 0.808 | 0.087 | 9.271*** |
| | Grade | | | | 0.445 | 0.074 | 6.028*** |
| | Ethnicity | | | | 0.001 | 0.160 | 0.004 |

Note: * P<0.05, ** P<0.01, *** P<0.001

**Table 5. Bootstrap-based serial mediation effects on life satisfaction.**

| Effect | Effect size | BootSE | BootLLCL | BootULCL | Proportion of total effect |
|---|---|---|---|---|---|
| Total effect | 0.253 | 0.005 | 0.244 | 0.263 | |
| Total direct effect | 0.183 | 0.005 | 0.173 | 0.194 | 72.40% |
| Total indirect effect | 0.070 | 0.004 | 0.063 | 0.077 | 27.60% |
| Exercise adherence→motor-skill competence→life satisfaction | 0.003 | 0.001 | 0.001 | 0.005 | 3.86% |
| Exercise adherence→health literacy→life satisfaction | 0.067 | 0.003 | 0.060 | 0.073 | 95.14% |
| Exercise adherence→motor-skill competence→health literacy→life satisfaction | 0.001 | 0.000 | 0.000 | 0.001 | 1.00% |

competence alone 0.003, and the serial chain (motor-skill competence→health literacy) 0.001. The model supports overall serial mediation, corroborating Hypotheses 2, 3, and 4.

The serial mediation path coefficients for life satisfaction are depicted in **Fig 2**.

## 4. Discussion

This study simultaneously examined the roles of motor-skill competence and health literacy in the effect of exercise adherence on the life satisfaction of university students. Using validated scales, the relationships stipulated in Hypotheses

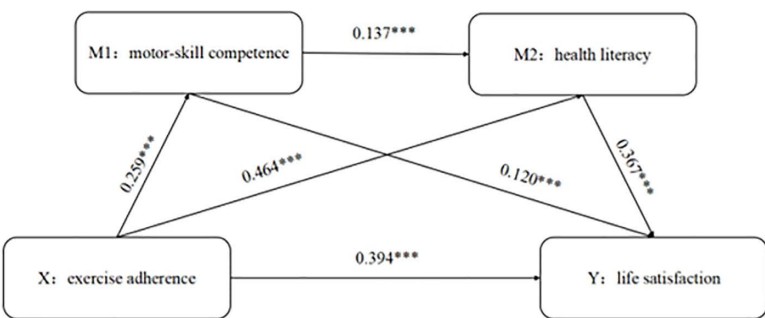

**Fig 2. Serial mediation model.**

1–4 were fully supported. Exercise adherence influences life satisfaction through a serial chain in which motor-skill competence and health literacy sequentially mediate the effect; in other words, the impact of adherence on satisfaction is channeled and modulated by these two factors. The Healthy China 2030 blueprint advocates "integrating health into all policies," highlighting mastery of motor skills and widespread health literacy as twin levers for national wellness [9]. Our findings indicate that both levers act as pivotal mediators in shaping university students' life satisfaction. This pattern illustrates a prevailing tension in college health promotion—policy momentum coexists with implementation lag, and cognitive gains outpace behavioral change—yet the positive trajectory is evident: motor skills are becoming more functional, health-literacy programs are emphasizing behavioral translation, and their synergistic contribution is increasingly recognized [50].

### 4.1. Relationship between exercise adherence and life satisfaction

The present study validates H1 by demonstrating that university students' exercise adherence is significantly and positively correlated with life satisfaction, and that adherence prospectively predicts higher satisfaction. This finding is consistent with self-determination theory, which posits that sustained exercise fulfills three basic psychological needs—autonomy, relatedness, and competence. By meeting these needs, exercise enhances perceived performance and, ultimately, life satisfaction [29]. Physiologically, habitual exercise triggers a sustained release of endorphins and serotonin while reducing cortisol levels, which directly elevates emotional tone [51–53]. This biochemical shift is complemented by the psychological experience of self-efficacy. The repeated experience of "I can do this" during exercise generalizes into a broader sense of mastery over daily life.

Moreover, regular physical activity serves as a long-term emotion-regulation strategy. It provides an outlet for negative emotions, and the post-exercise drop in core body temperature helps resynchronize circadian rhythms, deepen sleep, and further improve subjective evaluations of life satisfaction [54–56].

### 4.2. Mediating role of motor-skill competence

The study unequivocally demonstrates that motor-skill competence significantly mediates the relationship between exercise adherence and life satisfaction among Chinese university students, robustly supporting Hypothesis 2. This finding indicates that consistent exercise not only enhances students' physical capacity but also accelerates their mastery of movement patterns. This progress directly and significantly boosts their overall life satisfaction. The enhancement of motor-skill competence provides students with a strong sense of self-efficacy, enabling them to better integrate exercise into their daily routines and thereby improve their mental health and overall well-being.

When compared with findings in children, university students are more efficient in converting skill gains into psychological benefits. This efficiency can be attributed to their greater neuroplasticity and stronger institutional support [57,58].

These factors enable university students to more effectively translate improvements in motor skills into increased psychological satisfaction and life satisfaction.

The results of this study have significant implications for university physical education reforms. It is recommended that university physical education shift from the traditional "fitness-pass" systems to "skill-mastery" models. By using video feedback to visualize progress, universities can enhance students' motivation to continue exercising, diversify exercise formats, and ultimately significantly improve students' life satisfaction. This model not only enhances physical fitness but also, through the enhancement of motor-skill competence, provides lasting psychological benefits.

In summary, motor-skill competence significantly mediates the relationship between exercise adherence and life satisfaction among Chinese university students. This finding underscores the importance of emphasizing motor-skill development in university physical education. By adopting skill-mastery models and providing supportive environments, universities can effectively enhance students' life satisfaction and promote their holistic development.

### 4.3. Mediating role of health literacy

The mediation analysis reveals that health literacy significantly mediates the relationship between exercise adherence and life satisfaction among Chinese university students, thereby supporting Hypothesis 3. Specifically, exercise adherence positively predicts health literacy, which in turn positively predicts life satisfaction. These findings are consistent with prior research indicating that exercise adherence enhances health literacy [59] and that there is a strong association between health literacy and life satisfaction [60].

During sustained exercise, students with higher health literacy can design personalized training plans and accurately interpret bodily signals. This not only helps them adjust their exercise strategies more effectively but also prolongs their exercise adherence [61]. Unlike its role in disease management among older adults [62,63], health literacy in this context emphasizes the establishment of preventive health behaviors, which are crucial for meeting the independent living demands of university life. By enhancing health literacy, students can better manage their health and become more self-reliant in their daily lives.

As students' independent health capabilities improve through enhanced health literacy, they experience significant physical benefits, such as improved physical fitness and reduced risk of illness, as well as positive psychological changes, such as increased self-confidence and reduced stress. The increase in life satisfaction further reinforces their motivation to engage in healthy behaviors, creating a positive feedback loop. This highlights the critical role of health literacy in promoting both physical and psychological well-being and provides new insights for university physical education and health promotion programs.

In summary, health literacy significantly mediates the relationship between exercise adherence and life satisfaction among Chinese university students. This underscores the importance of health literacy in promoting overall well-being. Universities should prioritize the development of health literacy programs to support students' independent living skills and enhance their life satisfaction. By implementing these programs, universities can create a healthier and more positive learning environment, helping students better navigate the challenges of university life.

### 4.4. Serial mediating effect of motor-skill competence and health literacy

The findings confirm the serial mediation of motor-skill competence and health literacy, supporting Hypothesis 4. Sustained exercise elevates mood by releasing endorphins and improving sleep [51,64], creating an emotional milieu that facilitates skill practice. As skills advance, self-efficacy rises, generating heightened mastery and achievement during exercise [65]. Health literacy then translates these experiences into informed decisions: students who understand the exercise-health link plan training more scientifically, reduce injury risk, and maintain long-term activity, thereby enhancing quality of life [66].

Therefore, exercise adherence is associated with increased life satisfaction among university students and may indirectly influence life satisfaction through the serial pathway of motor-skill competence and health literacy.

## 4.5. Study limitations

This study is limited by its cross-sectional design, which precludes causal inferences. Single-time-point data collection does not allow for the observation of temporal sequences, making it difficult to establish causal relationships. Additionally, the single-item assessment of motor-skill competence may underestimate its true influence, warranting the development of a multidimensional scale in future research. Moreover, the sample consisted of Chinese university students, which limits the generalizability of the findings. The uneven distribution of participants across gender and grade levels may further affect the representativeness of the results. Caution should be exercised when applying these results to other populations due to potential cultural and educational differences.

## 5. Conclusion

Exercise adherence positively predicts university students' life satisfaction, motor-skill competence, and health literacy. Both motor-skill competence and health literacy independently mediate the relationship between exercise adherence and life satisfaction, and exercise adherence also exerts an indirect effect on life satisfaction through the serial mediation of motor-skill competence and health literacy.

## Acknowledgments

We sincerely thank all the staff and students from the participating schools and our co-operators for their assistance in data collection.

## Author contributions

**Conceptualization:** Shao-Shuai Ma, Jiaxian Geng.

**Data curation:** Jiaxian Geng.

**Formal analysis:** Ya-Xing Li.

**Funding acquisition:** Bo Li.

**Investigation:** Chen-Xi Li, Bo Li.

**Methodology:** Zhe Zhu.

**Project administration:** Jiaxian Geng.

**Resources:** Chen-Xi Li, Bo Li.

**Software:** Zhe Zhu.

**Supervision:** Sai Zhu.

**Validation:** Bo Li.

**Visualization:** Dongsheng Cai.

**Writing – original draft:** Dongsheng Cai, Chen-Xi Li, Bo Li.

**Writing – review & editing:** Bo Li.

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
