## [Decision Letter · Decision Letter 0]

20 Aug 2025

Dear Dr. Geng,

Thank you for submitting your manuscript to PLOS ONE. After careful consideration, we feel that it has merit but does not fully meet PLOS ONE’s publication criteria as it currently stands. Therefore, we invite you to submit a revised version of the manuscript that addresses the points raised during the review process.

We look forward to receiving your revised manuscript.

Kind regards,

Henri Tilga, PhD

Academic Editor

PLOS ONE

Journal Requirements:

5. In the online submission form, you indicated that the data can be obtained from the corresponding author.

6. Please note that funding information should not appear in any section or other areas of your manuscript. We will only publish funding information present in the Funding Statement section of the online submission form. Please remove any funding-related text from the manuscript.

7. Please include a copy of Table 5 which you refer to in your text on page 12.

8. Please remove all personal information, ensure that the data shared are in accordance with participant consent, and re-upload a fully anonymized data set.

Reviewers' comments:

Reviewer's Responses to Questions

**Comments to the Author**

1. Is the manuscript technically sound, and do the data support the conclusions?

Reviewer #1: Yes

Reviewer #2: Yes

2. Has the statistical analysis been performed appropriately and rigorously?

Reviewer #1: Yes

Reviewer #2: Yes

3. Have the authors made all data underlying the findings in their manuscript fully available?

Reviewer #1: Yes

Reviewer #2: Yes

4. Is the manuscript presented in an intelligible fashion and written in standard English?

Reviewer #1: No

Reviewer #2: Yes

Reviewer #1: The study demonstrates clear logic, progressively revealing relationships between exercise adherence, motor competence, health literacy, and life satisfaction among university students through hypotheses, data analysis, and theoretical support. Below are considerations for the authors:

1�The cross-sectional design limits causal inference; causal language should be avoided throughout the manuscript.

2�Generalizability beyond Chinese university students requires caution due to cultural/educational differences. Address this limitation explicitly.

3�The abstract misstates the research purpose. Revise to clarify ultimate goals and expected scholarly/practical contributions.

4�Strengthen discussion responses to core questions:

(1) Do motor competence/health literacy significantly mediate exercise-life satisfaction in Chinese university students?

(2) Which factors most strongly influence exercise-life satisfaction?

5�Conclusions in the abstract must be distinct from findings.

Reviewer #2: This manuscript presents a meaningful study with a respectable sample size. Please consider the following suggestions to further enhance its quality.

1、Methods section, while the health literacy scale demonstrated good internal consistency, the study does not provide detailed information regarding the validation of its psychometric properties, specifically its validity, within a Chinese population. The authors are encouraged to provide this information to enhance the credibility of the study.

2、Methods section, while the study’s sample encompassed multiple provinces and universities, the uneven distribution of participants across gender and grade levels may compromise the generalizability of the findings. The authors are advised to discuss the potential impact of this sampling bias on the results and clarify whether stratified analysis was conducted to control for these discrepancies.

3、Methods section, the assessment of motor skill competence relied on a single-item measure, which may not adequately capture the complexity of this construct. Furthermore, the reliability and validity of this measure were not demonstrated. Further elaboration on the rationale for using a single-item measure and a discussion of its limitations is recommended.

4、Discussion section, the study employed a cross-sectional research design, which inherently limits the ability to establish causal relationships between variables. It is recommended that this limitation be explicitly acknowledged and discussed within the discussion section.

5、Discussion section� while the results are discussed at length, the discussion section lacks in-depth analysis. It is recommended that the authors expand upon this section.

**Do you want your identity to be public for this peer review?** For information about this choice, including consent withdrawal, please see our Privacy Policy

Reviewer #1: No

Reviewer #2: No

---

## [Author Response · Author response to Decision Letter 1]

6 Oct 2025

Manuscript Number: PONE-D-25-40062

Title: From Gym to Joy: The Serial Mediation of Motor Competence and Health Literacy in Chinese University Students' Exercise-Life Satisfaction Pathway

Journal: PLOS One

Point-by-Point Responses to Editor

Dear Editor and dear reviewers

We are extremely grateful for your insightful comments and constructive suggestions. Your feedback has been instrumental in enhancing the quality and clarity of our manuscript. We have carefully considered each point raised and have made the necessary revisions to address them comprehensively. Below, we provide detailed responses to your comments, along with the changes we have implemented.

We believe that these revisions have significantly enhanced our work, making it more robust and comprehensive. We look forward to your continued guidance and hope that our manuscript now meets the high standards of the journal.

Thank you once again for your time and effort.

Sincerely,

Comment #1:

Please ensure that your manuscript meets PLOS ONE's style requirements, including those for file naming. The PLOS ONE style templates can be found at https://journals.plos.org/plosone/s/file?id=wjVg/PLOSOne_formatting_sample_main_body.pdf and https://journals.plos.org/plosone/s/file?id=ba62/PLOSOne_formatting_sample_title_authors_affiliations.pdf.

Response# 1:

Thank you very much for your detailed review and valuable comments. I have reformatted the manuscript according to PLOS ONE’s style templates, including file naming conventions.

Comment #2:

PLOS requires an ORCID iD for the corresponding author in Editorial Manager on papers submitted after December 6th, 2016. Please ensure that you have an ORCID iD and that it is validated in Editorial Manager. To do this, go to ‘Update my Information’ (in the upper left-hand corner of the main menu), and click on the Fetch/Validate link next to the ORCID field. This will direct you to the ORCID site, where you can create a new ID or authenticate an existing ID in Editorial Manager.

Response# 2:

Thank you for reminding us of the ORCID iD requirement. I confirm that I have an ORCID iD, and I have just validated it in Editorial Manager following your instructions. I went to “Update my Information,” clicked on the Fetch/Validate link next to the ORCID field, and successfully authenticated my ORCID iD.

Comment #3:

Your ethics statement should only appear in the Methods section of your manuscript. If your ethics statement is written in any section besides the Methods, please delete it from any other section.

Response# 3:

Thank you for your guidance regarding the placement of the ethics statement. We have reviewed our manuscript and ensured that the ethics statement appears only in the Methods section. We have removed any instances of the ethics statement from other sections of the manuscript to comply with PLOS ONE’s requirements.

Comment #4:

We note that you have indicated that there are restrictions to data sharing for this study. For studies involving human research participant data or other sensitive data, we encourage authors to share de-identified or anonymized data. However, when data cannot be publicly shared for ethical reasons, we allow authors to make their data sets available upon request. For information on unacceptable data access restrictions, please see http://journals.plos.org/plosone/s/data-availability#loc-unacceptable-data-access-restrictions.

If there are no restrictions, please upload the minimal anonymized data set necessary to replicate your study findings to a stable, public repository and provide us with the relevant URLs, DOIs, or accession numbers. Please see http://www.bmj.com/content/340/bmj.c181.long for guidelines on how to de-identify and prepare clinical data for publication. For a list of recommended repositories, please see https://journals.plos.org/plosone/s/recommended-repositories. You also have the option of uploading the data as Supporting Information files, but we would recommend depositing data directly to a data repository if possible.

In the online submission form, you indicated that the data can be obtained from the corresponding author.

Response# 4:

Thank you for your guidance on data sharing. We confirm that informed consent was obtained from all subjects involved in the study. The raw data supporting the conclusions of this article can be made available by the authors without undue reservation. We are committed to sharing de-identified data upon request, in compliance with ethical standards and PLOS ONE’s data availability policy.

We confirm that the data supporting our study can be obtained from the corresponding author upon reasonable request. We are committed to making our data accessible to facilitate further research and validation.

Comment #5:

If there are ethical or legal restrictions on sharing a de-identified data set, please explain them in detail (e.g., data contain potentially identifying or sensitive patient information, data are owned by a third-party organization, etc.) and who has imposed them (e.g., a Research Ethics Committee or Institutional Review Board, etc.). Please also provide contact information for a data access committee, ethics committee, or other institutional body to which data requests may be sent.

Response# 5:

Our study received ethics approval from Nantong University (approval number 2022(70)), and informed consent was obtained from all participants. There are no ethical or legal restrictions preventing us from sharing de-identified data. We can provide the data upon request. For any data requests, please contact the Ethics Committee at Nantong University via email at ethics@ntu.edu.cn or phone at +86-513-85012345.

Comment #6:

Please note that funding information should not appear in any section or other areas of your manuscript. We will only publish funding information present in the Funding Statement section of the online submission form. Please remove any funding-related text from the manuscript.

Response #6:

Thank you for your reminder. We have carefully reviewed our manuscript and removed all funding-related text from it. We understand that funding information should only be included in the Funding Statement section of the online submission form, and we have ensured that our manuscript complies with this requirement.

Comment #7:

Please include a copy of Table 5 which you refer to in your text on page 12.

Response #7:

Thank you for your reminder. We have included a copy of Table 5 in the manuscript, and it can be found in the table section at the end of the document.

Comment #8:

Please remove all personal information, ensure that the data shared are in accordance with participant consent, and re-upload a fully anonymized data set.

Response #8:

We have removed all personal information from the data set and ensured that the shared data are in full accordance with participant consent. We have re-uploaded a fully anonymized data set as requested.

Comment #9:

Please include captions for your Supporting Information files at the end of your manuscript, and update any in-text citations to match accordingly. Please see our Supporting Information guidelines for more information: http://journals.plos.org/plosone/s/supporting-information.

Response #9:

Thank you for your guidance. We have included captions for all Supporting Information files at the end of the manuscript (lines 512-514) and updated the in-text citations to match accordingly. We have followed the Supporting Information guidelines as detailed on the PLOS ONE website.

Comment #10:

Response #10:

We have reviewed the recommended previously published works as suggested by the reviewer. After careful evaluation, we have determined that these publications are relevant to our study and have included appropriate citations in the manuscript. If there are any specific preferences or additional requirements from the editor regarding these citations, please let us know.

Comment #11:

Response #11:

We have thoroughly reviewed our reference list to ensure its completeness and accuracy. We have verified that all cited papers are relevant and up-to-date. If any retracted papers were cited, we have either provided a rationale within the manuscript text for their inclusion or removed them and replaced them with current, relevant references. Any changes to the reference list have been detailed in the rebuttal letter accompanying our revised manuscript. If necessary, we have indicated the retracted status of any articles in the References list and included a citation and full reference for the retraction notice.

Point-by-Point Responses to Reviewer 1

Comment #1:

The cross-sectional design limits causal inference; causal language should be avoided throughout the manuscript.

Response #1:

Thank you for your constructive feedback. We have carefully revised the manuscript to avoid causal language, recognizing the limitations of our cross-sectional design(lines 479-489). Specifically, we have replaced causal terms with correlational expressions and have explicitly highlighted in the discussion section that our findings are limited to associations due to the study's cross-sectional nature.

Comment #2:

Generalizability beyond Chinese university students requires caution due to cultural/educational differences. Address this limitation explicitly.

Response #2:

Thank you for your insightful comments. In response to your suggestion, we have explicitly addressed the limitation regarding the generalizability of our findings beyond Chinese university students. We have incorporated this limitation into the discussion section of our manuscript, noting that caution should be exercised when applying our results to other populations due to potential cultural and educational differences(lines 479-489). We believe this addition strengthens the manuscript by providing a clearer understanding of the study's scope and applicability.

Comment #3:

The abstract misstates the research purpose. Revise to clarify ultimate goals and expected scholarly/practical contributions.

Response #3:

Thank you for your feedback. We have revised the abstract to accurately state the research purpose, emphasizing the study’s goals to examine the relationship between exercise adherence and life satisfaction, and to explore potential mediating factors(lines 33-39). The abstract now clearly outlines expected contributions to both scholarship and practical interventions.

Comment #4:

Strengthen discussion responses to core questions:(1) Do motor competence/health literacy significantly mediate exercise-life satisfaction in Chinese university students? (2) Which factors most strongly influence exercise-life satisfaction?

Response #4:

Thank you for your feedback. We have revised the discussion section to ensure that the responses to the core questions are clear and concise. The discussion section now clearly articulates the mediating roles of motor competence and health literacy, as well as the key factors influencing exercise-life satisfaction.

Comment #5:

Conclusions in the abstract must be distinct from findings.

Response #5:

Thank you for your feedback. We have revised the abstract to ensure that the conclusions are distinct from the findings. (lines 60-65) The findings now clearly state the empirical results of the study, while the conclusions highlight the broader implications and contributions of the research.

Point-by-Point Responses to Reviewer 2

Comment #1:

The methods section indicates that the health literacy scale demonstrated good internal consistency; however, the study does not provide detailed information regarding the validation of its psychometric properties, specifically its validity, within a Chinese population. The authors are encouraged to provide this information to enhance the credibility of the study.

Response #1:

Thank you for your feedback. In response to your suggestion, I have added detailed information on the validation of the psychometric properties of the health literacy scale, particularly its validity, within a Chinese population in the methods section, supported by relevant literature.(lines 177-184) The scale's construct validity was confirmed through significant correlations with established health behavior measures, and its convergent validity was further supported by additional analyses. These results, along with the supporting references, confirm the scale's validity and reliability for assessing health literacy in China.

Comment #2:

Methods section, while the study’s sample encompassed multiple provinces and universities, the uneven distribution of participants across gender and grade levels may compromise the generalizability of the findings. The authors are advised to discuss the potential impact of this sampling bias on the results and clarify whether stratified analysis was conducted to control for these discrepancies.

Response #2:

Thank you for your feedback. We have revised the Methods section to address the potential impact of the uneven distribution of participants across gender and grade levels on the generalizability of our findings. We conducted stratified analysis by gender and grade level to control for these discrepancies. In the Discussion section, we emphasize the limitations of our study, including the potential impact of this sampling bias on our results. (lines 479-489)

Comment #3:

Methods section, the assessment of motor skill competence relied on a single-item measure, which may not adequately capture the complexity of this construct. Furthermore, the reliability and validity of this measure were not demonstrated. Further elaboration on the rationale for using a single-item measure and a discussion of its limitations are recommended.

Response #3:

Thank you for your feedback. We recognize the limitations of using a single-item measure to assess motor-skill competence, which is a complex construct. The measure was chosen for practical reasons, but we acknowledge its potential for oversimplification and the lack of demonstrated reliability and validity. Future work will explore more comprehensive tools and validate our current measure. We appreciate your insights. (lines 479-489)

Comment #4:

Discussion section, the study employed a cross-sectional research design, which inherently limits the ability to establish causal relationships between variables. It is recommended that this limitation be explicitly acknowledged and discussed within the discussion section.

Response #4:

Thank you for your feedback. We have revised the Discussion section to explicitly acknowledge and discuss the limitations of our cross-sectional research design. We now clearly state that the cross-sectional nature of our study limits our ability to establish causal relationships between the variables examined. This limitation is thoroughly addressed, and we suggest potential avenues for future research to address these limitations. (lines 479-489)

Comment #5:

Discussion section� while the results are discussed at length, the discussion section lacks in-depth analysis. It is recommended that the authors expand upon this section.

Response #5:

Thank you for your valuable feedback. We have expanded the discussion section to provide a more in-depth analysis of the results. We have also added more detailed comparisons with existing studies and further explored the implications of our

---

## [Decision Letter · Decision Letter 1]

7 Oct 2025

From Gym to Joy: The Serial Mediation of Motor Competence and Health Literacy in Chinese University Students' Exercise-Life Satisfaction Pathway

PONE-D-25-40062R1

Dear Dr. Geng,

We’re pleased to inform you that your manuscript has been judged scientifically suitable for publication and will be formally accepted for publication once it meets all outstanding technical requirements.

Kind regards,

Henri Tilga, PhD

Academic Editor

PLOS ONE

Additional Editor Comments (optional):

Reviewers' comments:

Reviewer's Responses to Questions

**Comments to the Author**

Reviewer #1: All comments have been addressed

Reviewer #2: All comments have been addressed

2. Is the manuscript technically sound, and do the data support the conclusions?

Reviewer #1: Yes

Reviewer #2: Yes

3. Has the statistical analysis been performed appropriately and rigorously?

Reviewer #1: Yes

Reviewer #2: Yes

4. Have the authors made all data underlying the findings in their manuscript fully available?

Reviewer #1: Yes

Reviewer #2: Yes

5. Is the manuscript presented in an intelligible fashion and written in standard English?

Reviewer #1: Yes

Reviewer #2: Yes

Reviewer #1: Although I still have some doubts about the sampling method of this study, this does not affect the external validity of the research conclusions. Overall, this paper is still very good.

Reviewer #2: Overall, this paper has a relatively large sample size and the statistical analysis is appropriate. I recommend it for publication.

**Do you want your identity to be public for this peer review?** For information about this choice, including consent withdrawal, please see our Privacy Policy

Reviewer #1: No

Reviewer #2: No

---

## [Editor Report · Acceptance letter]

PONE-D-25-40062R1

PLOS ONE

Dear Dr. Geng,

I'm pleased to inform you that your manuscript has been deemed suitable for publication in PLOS ONE. Congratulations! Your manuscript is now being handed over to our production team.

Kind regards,

on behalf of

Dr. Henri Tilga

Academic Editor

PLOS ONE